# Spatial-Temporal-Decoupled Meta Learning for Dynamic Node Attribute Forecasting

## Abstract

Dynamic graph forecasting has become increasingly important in various domains, such as social networks and transportation systems. While dynamic graph neural networks (GNNs) have shown promise in predicting future node attributes, they often fail to capture the complex spatial-temporal interactions between nodes, limiting their performance. In this paper, we introduce STDMD, a novel dynamic GNN model that incorporates meta spatial-temporal decoupling to effectively capture both spatial and temporal dependencies in node attributes. By leveraging meta-learning, STDMD adapts to the evolving spatiotemporal patterns of node data, improving the accuracy and robustness of predictions. Specifically, our model dynamically refines spatial and temporal representations through an iterative meta-optimization process, allowing for more effective learning of dynamic node interactions. Furthermore, STDMD is designed to generalize across different dynamic graph structures, making it highly scalable and adaptable to real-world applications. Experimental results on real-world datasets demonstrate that STDMD outperforms state-of-the-art baselines, showcasing its ability to model dynamic node attributes with greater precision and robustness.

## 1 Introduction

Node attribute forecasting plays a critical role across a variety of domains, including traffic flow prediction, disease transmission monitoring, and web page access detection. Accurate and reliable forecasting of dynamic node attributes is essential for controlling the development of dynamic graphs and preventing potentially catastrophic consequences. For example, by monitoring traffic flow on road maps, we can provide real-time guidance to vehicles, reduce congestion, and optimize traffic management. Similarly, in healthcare, tracking the number of people infected with a disease allows for predicting infection rates and identifying potential transmission paths, which is crucial for controlling outbreaks. Therefore, developing methods that accurately forecast node attributes and capture the complex variations in dynamic graphs is of paramount importance.

Recent advancements in graph neural networks (GNNs) have significantly enhanced node representation learning, with many models integrating GNNs with time series networks (RNN, Transformer) Jiang et al. (2021); Li et al. (2018; 2023a); Liu et al. (2023); Chen et al. (2025); Weng et al. (2023); Fan et al. (2025) to capture the temporal dynamics of evolving graphs. These hybrid approaches have demonstrated strong potential in modeling sequential dependencies within dynamic graphs. However, they often suffer from an issue denoted as the dynamic graph mirage: **i)** ***the difficulty of capturing complex spatial-temporal dependencies.*** **ii)** ***the inability to adapt to evolving dynamic graph patterns.***

For **i)**, Conventional models often treat spatial and temporal information independently—focusing either on structural connectivity between nodes or temporal changes in node attributes—while neglecting their intricate interdependence. As illustrated in Figure 1, at time step $T_1$, disease transmission follows local mobility patterns (e.g., $A \to C$, $B \to E$) via direct regional connections. However, by time step $T_2$, the infection unexpectedly spreads to regions $C$, $D$, $E$, and even distant $G$, driven by long-range transport effects. This mismatch stems from the dynamic nature of spatial structures and the irregularity of temporal variations. Failing to model their joint influence leads to limited adaptability and suboptimal forecasting, particularly in rapidly evolving environments.

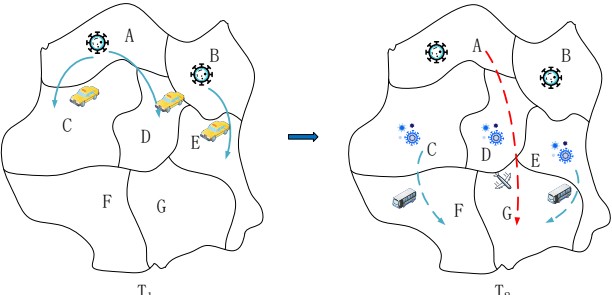

Figure 1: The virus originates from sources A and B. Traditional models mainly rely on direct spatial connections (solid blue arrows), focusing on vehicle movement and local spread. Yet, sudden outbreaks in regions C, D, and E suggest the presence of hidden spatiotemporal factors that such models fail to capture. Moreover, long-range transmission through public transport and airborne spread to region G (red dashed arrow) further expose the limitations of static approaches.

For **ii**), In real-world dynamic graphs, node attributes evolve irregularly under the influence of complex spatial-temporal factors—such as cascading failures in transportation, fluctuating infection rates, or shifting user behaviors. As shown in Figure 1, although traditional models anticipate local transmission based on static adjacency (solid blue arrows), the infection spreads non-linearly to regions like $C$, $D$, and $E$, which are not directly connected to the original sources $A$ and $B$. This spread is driven by latent influences like public transport and air travel, which static models fail to account for. Without mechanisms to adapt to evolving graph topologies and hidden dependencies, traditional forecasting approaches often mispredict future states in highly dynamic settings.

These challenges arise because conventional models typically focus on either the spatial structure of nodes or their temporal dependencies in isolation, failing to capture their intricate interdependencies over long time spans. Node attribute changes are inherently irregular and influenced by a wide array of spatial and temporal factors, such as cascading effects in transportation systems, fluctuating infection rates in epidemiology, or evolving user behavior in online platforms. Consequently, current approaches struggle to generalize effectively across diverse and highly dynamic environments. This limitation hinders their predictive accuracy and restricts their practical deployment in real-world applications that require adaptability to rapidly changing conditions, such as intelligent traffic control, epidemic forecasting, and financial market analysis.

To address these challenges, we propose STDMD, a novel dynamic GNN model that integrates meta-learning to enhance GNN for node attribute forecasting. Specifically, we design three key modules: the encoding layer, task construction, and meta optimization. The encoding layer leverages a GNN combined with a gating mechanism to generate robust node embeddings that capture both spatial and temporal dependencies. In the task construction module, we create support sets and query sets for both spatial and temporal tasks, enabling the model to effectively learn from different types of graph interactions. Finally, the meta optimization step applies meta-learning to dynamically update both the spatial and temporal tasks, as well as the model parameters, to better capture the evolving patterns in the dynamic graph. This process allows STDMD to significantly improve its ability to predict future node attributes by more accurately modeling the spatiotemporal dependencies inherent in dynamic graphs. In summary, our key contribution is threefold:

- We propose STDMD, a novel dynamic graph neural network model that integrates meta spatial-temporal graph learning to effectively capture the complex spatiotemporal dependencies in dynamic node attributes, improving the forecasting accuracy.

- We design a meta-learning framework that dynamically updates both spatial and temporal tasks, enhancing the model's ability to adapt to varying patterns of node attribute evolution in dynamic graphs.

- We demonstrate the effectiveness of STDMD through extensive experiments on multiple real-world datasets. Our results show that STDMD outperforms state-of-the-art baselines, both in terms of quantitative forecasting performance and its ability to capture meaningful spatiotemporal interactions.

## 2 RELATED WORK

### 2.1 SPATIOTEMPORAL FORECASTING

Spatiotemporal forecasting Jiang et al. (2021); Dong et al. (2024); Cini et al. (2023) aims to predict future trends by modeling historical spatiotemporal patterns, essential for dynamic node attribute learning. Recent works combine graph neural networks (GNNs) with recurrent architectures to jointly capture spatial and temporal dependencies. For example, Graph WaveNet Wu et al. (2019) uses wavelet-based graph convolution for dynamic relationship modeling, while StemGNN Cao et al. (2020) leverages DFT and GFT to extract frequency-domain features. MTGNN Wu et al. (2020) integrates graph learning and temporal convolution to learn patterns and structures simultaneously. ST-Norm Deng et al. (2021) applies spatial and temporal normalization to disentangle high-frequency components in complex systems. Pre-training approaches like GPT-ST Li et al. (2023b) and STD-MAE Gao et al. (2024) further enhance performance via masked autoencoders. Transformer-based models Jiang et al. (2023a); Liu et al. (2023) also show strong capabilities in long-range dependency modeling. However, most existing methods focus on macro-level spatiotemporal features, overlooking fine-grained dynamics in evolving node attributes.

### 2.2 DYNAMIC GRAPH META LEARNING

Dynamic graph learning addresses scenarios where both graph structures and node attributes evolve over time. Traditional models often struggle to capture such dynamics effectively. Meta-learning Hospedales et al. (2022); Son et al. (2025); Wang et al. (2025) provides a promising solution by enabling rapid adaptation to new tasks with limited data, making it well-suited for dynamic graphs. Recent studies have integrated meta-learning into dynamic GNNs to improve adaptability and predictive accuracy. For example, MegaCRN Jiang et al. (2023b) introduces a MetaNode Bank within a graph convolutional recurrent encoder-decoder framework. MetaDGE Mao et al. (2024) leverages Model-Agnostic Meta-Learning to learn dynamic graph embeddings. DMetaGCRN Guo et al. (2025) proposes a meta-graph generator and a dynamic meta-graph recurrent unit to jointly model spatial and temporal dependencies.

## 3 PROBLEM DEFINITION

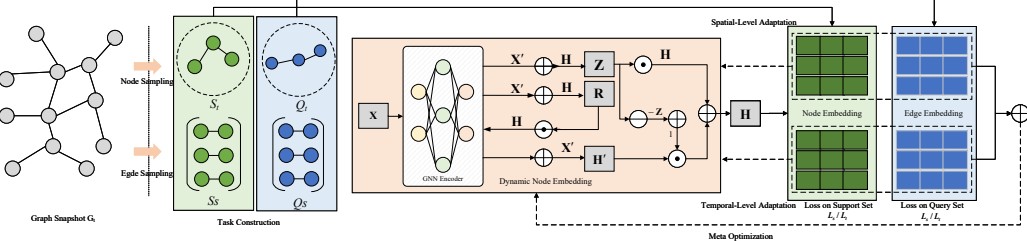

Figure 2: Given a dynamic graph snapshot $G_t$, STDMD constructs spatiotemporal tasks by sampling nodes and edges to form support and query sets. A GNN encoder processes input features $X$ to produce dynamic node embeddings, integrating spatial and temporal dependencies via gating mechanisms ($\mathbf{Z}$, $\mathbf{R}$, $\mathbf{H}'$). Meta-optimization is performed at both spatial and temporal levels, enabling joint adaptation of node and edge embeddings. This design ensures robust generalization for forecasting over evolving graph structures.

**Dynamic Graph.** Given a dynamic graph $\mathcal{G} = (\mathcal{V}, \mathcal{E}, \mathcal{X})$, where $\mathcal{V}$ represents the set of nodes, $\mathcal{E}$ denotes the set of edges, and $\mathcal{X} = \{\mathbf{x}_t \mid t = 1, \dots, T\}$ is the sequence of node attribute matrices over time, the goal is to predict the future attributes of nodes.

**Node Attribute Forecasting.** Formally, the problem can be defined as learning a function $f : (\mathcal{G}_t, \mathbf{x}_t) \to \mathbf{x}_{t+1}$, where $\mathcal{G}_t = (\mathcal{V}, \mathcal{E}_t)$ represents the graph structure at time $t$, $\mathbf{x}_t \in \mathbb{R}^{|\mathcal{V}| \times d}$ is the node attribute matrix at time $t$, $\mathbf{x}_{t+1}$ is the predicted node attribute matrix at time $t + 1$.

# 4 METHODOLOGY

Our model consists of three modules, namely encoding layer, task construction, and meta-learning.

## 4.1 TASK CONSTRUCTION

In this section, we describe the construction of spatial and temporal tasks in dynamic graphs, which are essential for training a meta-learning framework capable of capturing evolving spatiotemporal dependencies. By partitioning the graph into support and query sets, the model learns to generalize across different spatial structures and temporal dynamics, enabling fast adaptation to unseen graph patterns.

Dynamic graphs exhibit both spatial correlations and temporal dependencies, where spatial relationships determine how nodes interact within each snapshot, while temporal dependencies define how node attributes evolve over time. To effectively learn from these dynamic interactions, we construct two types of tasks:

- Spatial tasks ($\mathcal{T}_s$): These focus on learning relationships between nodes and their neighbors by sampling edges.

- Temporal tasks ($\mathcal{T}_t$): These focus on modeling the evolution of node attributes by sampling nodes over different time steps.

Each task consists of a support set (used for adaptation) and a query set (used for evaluation), following the standard meta-learning paradigm. While we decouple spatial and temporal tasks for meta-learning, these two components are not treated in isolation. Through the joint meta-optimization step, spatial-level and temporal-level adaptations are optimized simultaneously, enabling implicit modeling of spatiotemporal interactions. This design ensures that cross-dependencies are captured without introducing the instability often observed in monolithic spatiotemporal objectives.

**Spatial Task Construction.** Consider a dynamic graph $\mathcal{G} = (\mathcal{V}, \mathcal{E}, \mathcal{X}, \mathcal{Z})$, where $\mathcal{V}$ is the set of nodes, $\mathcal{E}$ is the set of edges, $\mathcal{X}$ represents node attributes, and $\mathcal{Z}$ represents the temporal states of the graph. To capture spatial dependencies, we define a spatial task $\mathcal{T}_s$ as a pair of mutually exclusive subsets of edges:

$$\mathcal{T}_s = (\mathcal{S}_s = \{(u,v) \in \mathcal{E}\}, \mathcal{Q}_s = \{(p,q) \in \mathcal{E}\}) \\ s.t. \mathcal{S}_s \cap \mathcal{Q}_s = \emptyset, \tag{1}$$

$\mathcal{S}_s$ (Support Set): A subset of edges sampled from $\mathcal{E}$, used to train the model on spatial structures. $\mathcal{Q}_s$ (Query Set): Another subset of edges sampled from $\mathcal{E}$, but disjoint from the support set, used to evaluate the model's ability to generalize to unseen spatial patterns. Each sampled edge $(u,v) \in \mathcal{S}_s$ contains nodes $u$ and $v$, and the model learns to predict the relationship between them. The key challenge is that spatial relationships are dynamic, meaning that new edges may form, or existing edges may disappear over time. By training on $\mathcal{S}_s$ and testing on $\mathcal{Q}_s$, the model learns to generalize spatial interactions beyond the observed connections.

Example: In a transportation network, roads (edges) connect different locations (nodes). Some roads may experience congestion, temporary closures, or new route openings. The spatial task enables the model to predict new road connections and adapt to network changes.

**Temporal Task Construction.** While spatial relationships describe how nodes interact within a snapshot, temporal dependencies define how node attributes evolve over time. To model temporal evolution, we construct a temporal task $\mathcal{T}_t$, where nodes are sampled at different time steps:

$$\mathcal{T}_t = (\mathcal{S}_t = \{v_i \in \mathcal{V}\}, \mathcal{Q}_t = \{v_j \in \mathcal{V}\}), s.t. \mathcal{S}_t \cap \mathcal{Q}_t = \emptyset. \tag{2}$$

$\mathcal{S}_t$ (Support Set): A set of nodes sampled from $\mathcal{V}$ at earlier time steps, used for training the model to learn temporal patterns. $\mathcal{Q}_t$ (Query Set): A set of nodes sampled at later time steps, used to evaluate the model's ability to predict future node attributes. By using disjoint sets $\mathcal{S}_t$ and $\mathcal{Q}_t$, we ensure that the model learns temporal representations from past observations and generalizes to future time steps.

Example: In an epidemic forecasting scenario, the number of infections (node attributes) changes over time. The model trained on past infections (support set) should be able to predict future outbreaks (query set) based on temporal patterns.

By separately defining spatial and temporal tasks, we enable meta-learning to effectively decouple spatial and temporal dependencies, allowing the model to adapt efficiently to unseen dynamic graph structures. The support-query task split ensures that the model learns generalizable patterns, making it robust to real-world dynamic environments where both spatial structures and temporal behaviors continuously evolve.

## 4.2 DYNAMIC NODE EMBEDDING

This section describes the method for encoding the dynamic graph and generating high-quality node embeddings. The encoding process is divided into two stages: applying the GCN layer to capture the spatial structure and using a gating mechanism to enhance the temporal dynamics of the node embeddings.

To begin with, the GCN layer encodes the structural information of the graph. We define the operation of the GCN layer as follows:

$$\mathbf{X}' := GCN(\mathbf{X}) = \hat{\mathbf{D}}^{-1/2}\hat{\mathbf{A}}\hat{\mathbf{D}}^{-1/2}\mathbf{X}\boldsymbol{\Theta}, \tag{3}$$

where $\hat{\mathbf{A}} = \mathbf{A} + \mathbf{I}$ denotes the adjacency matrix of the graph augmented with self-loops. Here, $\mathbf{A}$ is the original adjacency matrix, and $\mathbf{I}$ is the identity matrix. The diagonal degree matrix $\hat{\mathbf{D}}$ is calculated as $\hat{D}_{ii} = \sum_{j=0} \hat{A}_{ij}$. $\mathbf{X}$ represents the initial node attribute matrix, which contains the features of all nodes. $\boldsymbol{\Theta}$ represents the learnable weights of a multi-layer perceptron (MLP) that transforms the node features. This operation computes $\mathbf{X}'$, the node embeddings that incorporate the graph's structural information and prepare the input for the subsequent temporal update process.

Next, to capture the temporal evolution of the dynamic graph, we employ a gating mechanism inspired by LSTM. This mechanism enables the model to update node embeddings adaptively based on historical states. The process consists of several steps: The update gate regulates the extent to which the current node information $\mathbf{X}'$ and the historical state $\mathbf{H}$ are incorporated into the new embedding:

$$\mathbf{Z} = \sigma(\mathbf{X}' + \mathbf{H}), \tag{4}$$

where $\mathbf{H}$ is initialized as a zero matrix of the same shape as $\mathbf{X}$, and $\sigma$ denotes an activation function such as ReLU or Sigmoid. The matrix $\mathbf{Z}$ thus determines the degree to which past information is preserved.

The reset gate controls how much of the historical state is forgotten when computing the current embedding:

$$\mathbf{R} = \sigma(\mathbf{X}' + \mathbf{H}), \tag{5}$$

yielding the reset gate matrix $\mathbf{R}$.

Based on this gating mechanism, the candidate state $\mathbf{H}'$ is derived by integrating the gated historical information with the current graph structure:

$$\mathbf{H}' = \sigma\big(\mathbf{X}' + GCN(\mathbf{H} \cdot \mathbf{R})\big), \tag{6}$$

where $\mathbf{H} \cdot \mathbf{R}$ applies the reset gate to the historical embeddings, and the GCN layer propagates spatial dependencies.

Finally, the output embedding is obtained as a weighted combination of the previous state $\mathbf{H}$ and the candidate state $\mathbf{H}'$, modulated by the update gate $\mathbf{Z}$:

$$\mathbf{H} = \mathbf{Z} \cdot \mathbf{H} + (1 - \mathbf{Z}) \cdot \mathbf{H}'. \tag{7}$$

This adaptive update enables the embeddings to capture both spatial and temporal information, while the initial state $\mathbf{H}$ is set to a zero matrix to ensure consistency with the dimensionality of $\mathbf{X}$. This design effectively combines the structural information captured by GCN with temporal dynamics through the gating mechanism. The activation functions (e.g., ReLU or Sigmoid) ensure non-linear transformations, enabling the model to learn complex patterns in dynamic graphs. As a result, this encoding layer serves as the backbone for extracting high-quality node embeddings that form the basis for downstream tasks.

### 4.3 META OPTIMIZATION.

Meta-learning intends to learn a form of general knowledge across similar learning tasks so that the learned knowledge can be quickly adapted to new tasks Peng (2020). In our work, we intend to explore the spatial interaction and temporal change rules of nodes in the dynamic graph and predict the future value of node attributes through meta-learning under limited current information.

**Spatial-Level Adaptation.** To effectively capture the spatial connections between nodes in a dynamic graph, we design a spatial task that focuses on modeling the relationships among nodes. This task is implemented using a contrastive learning approach, which encourages embeddings of connected nodes to be similar while ensuring embeddings of unconnected nodes remain distinct. The spatial task loss is formulated as:

$$\mathcal{L}_s(\omega, \mathcal{S}_s) = \sum_{(u,v) \in \mathcal{S}_s} -\ln \sigma(\mathbf{h}_u \mathbf{h}_v^T) - \ln \sigma(-\mathbf{h}_u \mathbf{h}_{v'}^T), \tag{8}$$

where $v'$ is a negative node sample that is not linked with $u$. The learnable parameters $\omega$ (i.e., $\boldsymbol{\Theta}$) for both space and temporal tasks represent meta-knowledge.

**Temporal-Level Adaption.** Then, we design a temporal task to simulate the change happening on node attributes, formulated as

$$\mathcal{L}_t(\omega, \mathcal{S}_t) = \frac{1}{|\mathcal{S}_t|} \sum_{v_i \in \mathcal{S}_t} (\sigma(\mathbf{h}_{v_i}) - y)^2, \tag{9}$$

where $y$ is the target value of the node attribute in the future. Besides, we combine the spatial and temporal tasks to provide significant gain for capturing inherent evolving patterns in the dynamic graph. With the space and temporal adaptations on the query set, we can obtain a more scalable dynamic GCN model. The adaptation loss is formulated as

$$\theta \leftarrow \theta - \gamma \nabla_\theta \left( \mathcal{L}_s \left( \mathcal{Q}_s \right) + \beta \mathcal{L}_t \left( \mathcal{Q}_t \right) \right), \tag{10}$$

where the parameters $\theta$ (i.e., $\boldsymbol{\Theta}$) is optimized to quickly adapt the model to changes on node attributes in the dynamic graph. The $\gamma$ is the learning rate of the model, and $\beta$ denotes the balance coefficient between two task losses.

The proposed meta-optimization framework offers several key advantages. It enables the model to generalize effectively to new and unseen scenarios in dynamic graphs by leveraging meta-knowledge from both spatial and temporal tasks, enhances scalability through joint spatial-temporal optimization to adapt to diverse patterns of node attribute evolution, and facilitates rapid adaptation to changes in node states with minimal computational overhead using gradient-based meta-optimization. This integrated approach allows the dynamic GCN model to efficiently capture and predict the evolution of node attributes in complex dynamic graph environments.

## 5 EXPERIMENT

### 5.1 DATASETS

We conduct extensive experiments on three real-world datasets to validate the effectiveness and robustness of our proposed approach. WikiMaths Rozemberczki et al. (2021), which captures user interactions and content updates on a collaborative platform between March 16th 2019 and March 15th 2021 which results in 731 periods. EnglandCovid Panagopoulos et al. (2021), a dataset reflecting the spatiotemporal dynamics of COVID-19 case distributions across different regions in England from 3 March to 12 of May. PedalMe Rozemberczki et al. (2021), which tracks the operational and mobility data of a sustainable urban logistics service in London between 2020 and 2021. Each dataset presents unique challenges in terms of dynamic graph structure, temporal dependencies, and heterogeneous features, enabling a comprehensive evaluation of our method's performance across diverse scenarios. The detailed information of datasets are summarized in Table 1.

### 5.2 BASELINES

To comprehensively evaluate the effectiveness of our proposed method, we benchmark it against several state-of-the-art models. The selected baselines include GCN Kipf & Welling (2017), GConvGRU Seo et al. (2018), and GConvLSTM Chen et al. (2022), which primarily focus on spatial and

Table 1: Summary of the datasets used in our experiments.

| Dataset | # Nodes | # Edges (avg.) | # Time Steps |
|---------|---------|----------------|--------------|
| WikiMaths | 1,068 | 27,079 | 722 |
| EnglandCovid | 129 | 1,743 | 52 |
| PedalMe | 15 | 225 | 30 |

sequential dependencies. We also consider spatiotemporal forecasting methods, including DCRNN Li et al. (2018), STGCN Yu et al. (2018), and STSGCN Song et al. (2020), which explicitly integrate spatial and temporal information. Furthermore, dynamic graph models such as EvolveGCNO Pareja et al. (2020) and TGCN Zhao et al. (2020) are evaluated to assess their adaptability to evolving graph structures. In addition, we include recent transformer-based approaches such as STAEformer Liu et al. (2023), PDFormer Li et al. (2023a), and STWave Fang et al. (2023). We compare it with recent meta-learning-enhanced dynamic graph models, including MegaCRN Jiang et al. (2023b), MetaDGE Mao et al. (2024) and DMetaGCRN Guo et al. (2025).

## 5.3 SETTINGS

We configure the experimental settings as follows. For training Configuration, the training process spans 100 epochs, with the dataset split into training, validation, and test sets according to a 8:1:1 ratio. For each task, we randomly sample 100 edges/nodes for the support set ($k_{spt}$) and 100 edges/nodes for the query set ($k_{qry}$). For model architecture, the model takes an input dimension of 4 and uses a hidden dimension of 32. A dropout rate of 0.5 is applied to prevent overfitting. For optimization parameters, The meta-learning rate and inner update learning rate both set to 0.01. The number of update steps for both spatial tasks and temporal tasks is set to 1. Balance coefficient $\beta$ is set to 0.5. We evaluate the test performance with the mean squared errors (MSE), root mean squared error (RMSE), and mean absolute percentage error (MAPE(%)). (Our environment: CPU: Intel(R) Xeon(R) Silver 4210 CPU @ 2.20GHz, GPU: NVIDIA RTX 4090@24GB, Memory: 128GB. The implementation of our model and all baselines are based on Pytorch 1.9.0 and Python 3.9)

Table 2: Forecasting errors evaluated by MSE, MAPE, and RMSE on three real-world datasets. Bold values indicate the best results. * and ** denote p-value < 0.05 and 0.01 respectively in paired t-test against the second-best method.

| Models | MSE | | | MAPE | | | RMSE | | |
|--------|-----|--|--|------|--|--|------|--|--|
| | WikiMaths | EnglandCovid | PedalMe | WikiMaths | EnglandCovid | PedalMe | WikiMaths | EnglandCovid | PedalMe |
| GCN | 0.8211 | 0.9723 | 1.1512 | 9.12% | 12.34% | 15.45% | 0.9062 | 0.9860 | 1.0730 |
| GConvGRU | 0.7931 | 0.9412 | 1.2016 | 8.89% | 11.78% | 14.98% | 0.8906 | 0.9702 | 1.0962 |
| GConvLSTM | 0.7991 | 0.9541 | 1.2141 | 8.93% | 11.92% | 15.12% | 0.8940 | 0.9767 | 1.1019 |
| DCRNN | 0.8061 | 0.8323 | 1.2213 | 8.95% | 11.35% | 15.21% | 0.8980 | 0.9123 | 1.1042 |
| EvolveGCNO | 0.7783 | 0.9793 | 1.2013 | 8.80% | 12.48% | 15.10% | 0.8822 | 0.9896 | 1.0960 |
| TGCN | 0.7875 | 0.8587 | 1.2515 | 8.85% | 11.69% | 15.65% | 0.8874 | 0.9266 | 1.1186 |
| STGCN | 0.7823 | 0.8432 | 1.2015 | 8.79% | 11.40% | 15.00% | 0.8845 | 0.9182 | 1.0961 |
| STSGCN | 0.7795 | 0.8211 | 1.1753 | 8.76% | 11.23% | 14.85% | 0.8830 | 0.9051 | 1.0841 |
| STWave | 0.7704 | 0.7983 | 1.1421 | 8.65% | 10.98% | 14.62% | 0.8776 | 0.8935 | 1.0683 |
| STAEformer | 0.7875 | 0.8587 | 1.2515 | 8.85% | 11.69% | 15.65% | 0.8874 | 0.9266 | 1.1186 |
| PDFormer | 0.7875 | 0.8587 | 1.2515 | 8.85% | 11.69% | 15.65% | 0.8874 | 0.9266 | 1.1186 |
| MegaCRN | 0.7721 | 0.7453 | 1.1135 | 8.58% | 9.85% | 14.05% | 0.8791 | 0.8632 | 1.0554 |
| MetaDGE | 0.7672 | 0.6987 | 1.0896 | 8.53% | 9.12% | 13.92% | 0.8755 | 0.8365 | 1.0433 |
| DMetaGCRN | 0.7625 | 0.6451 | 1.0543 | 8.48% | 8.77% | 13.68% | 0.8728 | 0.8036 | 1.0271 |
| **STDMD** | **0.7650**\*\* | **0.5411**\*\* | **1.0106**\* | **8.50%**\* | **8.32%**\*\* | **13.45%**\* | **0.8747**\* | **0.7354**\*\* | **1.0053**\* |

## 5.4 OVERALL PERFORMANCE

As summarized in Table 2, STDMD consistently delivers the best forecasting accuracy across all three datasets, significantly outperforming both classical spatiotemporal models (e.g., DCRNN, STGCN, STWave) and more recent meta-learning based approaches (e.g., MegaCRN, MetaDGE, DMetaGCRN). In particular, STDMD achieves the lowest errors on EnglandCovid and PedalMe, two datasets characterized by highly volatile temporal dynamics and heterogeneous spatial interactions, underscoring its robustness to complex and noisy real-world conditions. Statistical signifi-

cance tests further confirm the superiority of STDMD, with most improvements achieving $p < 0.05$ or $p < 0.01$ when compared against the second-best methods.

The performance gain of STDMD arises from three core design choices. First, the GCN-based encoder with gating generates expressive spatiotemporal embeddings by dynamically refining spatial features, allowing the model to capture both stable structures and transient variations. Second, the unified meta-learning framework jointly optimizes spatial and temporal objectives through a support–query mechanism, which improves generalization to unseen temporal fluctuations and structural perturbations. Third, the dual-level meta-optimization strategy provides adaptive parameter updates across tasks, balancing accuracy with training efficiency and avoiding the overfitting commonly observed in prior methods.

### 5.5 ABLATION STUDY

To assess the contribution of individual components in our framework, we conduct a systematic ablation study with two model variants. **S-STDMD** removes the temporal task ($\mathcal{L}_t$) and optimizes only with the spatial task ($\mathcal{L}_s$), thereby testing the role of temporal dynamics in dynamic graphs. Conversely, **T-STDMD** removes the spatial task ($\mathcal{L}_s$) and relies solely on the temporal task ($\mathcal{L}_t$), highlighting the importance of spatial dependencies. As shown in Figure 3, STDMD consistently achieves the best results across all datasets, demonstrating its effectiveness in capturing and forecasting dynamic graph patterns. Both S-STDMD and T-STDMD yield competitive but inferior performance, confirming the necessity of jointly modeling spatial and temporal components.

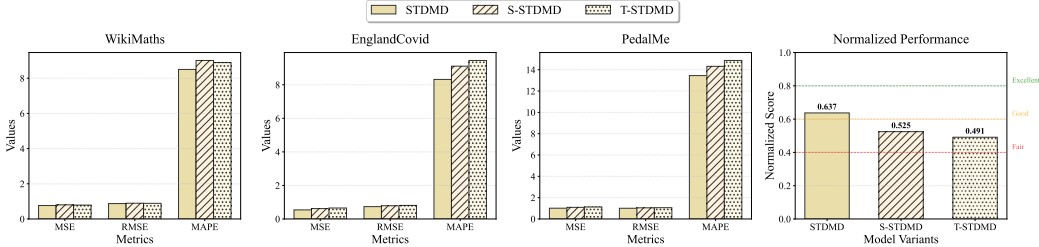

Figure 3: Ablation study on WikiMaths, EnglandCovid, PedalMe.

### 5.6 HYPERPARAMETERS ANALYSIS

We investigate the effects and sensitivity of our proposed STDMD on WikiMaths dataset to different hyperparameters, including meta learning rate $\alpha$ and update learning rate $\gamma$, number of support $k_{spt}$ and query samples $k_{qry}$, hidden dimension size $d$ and balance coefficient $\beta$. Figure 4 shows the influence of key parameters on model performance. For learning rates, both meta learning rate and update learning rate demonstrate a U-shaped relationship with MSE, where extremely low or high values degrade performance. The optimal range, observed around $\alpha$=0.01 and $\gamma$=0.01, ensures efficient optimization and stable convergence. Regarding sample sizes, increasing $k_{spt}$ and $k_{qry}$ improves performance up to a point, with the best results at $k_{spt}$=100 and $k_{qry}$=100, balancing generalization and computational cost. Lastly, the hidden dimension size $d$ exhibits diminishing returns. Performance stabilizes with dimensions above 128. A range of 32-128 is recommended for good accuracy while avoiding overfitting or unnecessary complexity. The balance coefficient ($\beta$) determines the trade-off between spatial and temporal dependencies. Our results show that $\beta = 0.5$ achieves the lowest MSE, indicating an optimal balance. Higher or lower values lead to performance degradation. These findings offer practical guidelines for hyperparameter tuning to enhance the model's prediction performance on the WikiMaths dataset.

### 5.7 EFFICIENCY TEST

Table 3 reports the efficiency results across WikiMaths, EnglandCovid, and PedalMe. STDMD achieves the lowest training times (110s, 90s, and 100s) and inference times (4.2s, 4.0s, and 4.1s), demonstrating clear computational advantages. Its time complexity is approximately $\mathcal{O}(EF + NF^2 + k_{spt}F^2 + k_{qry}F^2)$, where $E$, $N$, and $F$ denote the number of edges, nodes, and

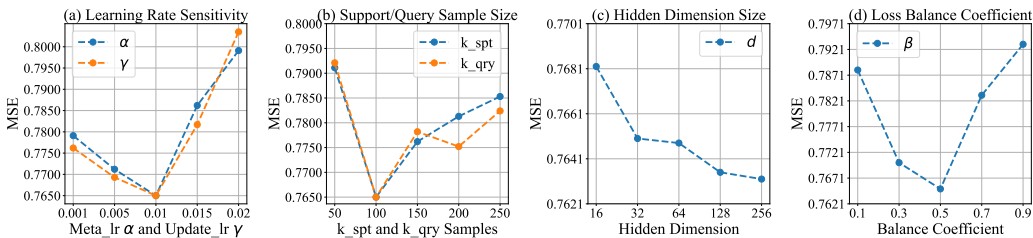

Figure 4: Hyperparameters Analysis on WikiMaths Dataset (MSE).

Table 3: Efficiency on three datasets (T.=Training, I.=Inference).

| Model | WikiMaths | | EnglandCovid | | PedalMe | |
|---|---|---|---|---|---|---|
| | **T.** | **I.** | **T.** | **I.** | **T.** | **I.** |
| GConvGRU | 145s | 6.3s | 120s | 6.0s | 135s | 6.2s |
| DCRNN | 175s | 7.0s | 140s | 6.9s | 160s | 7.1s |
| EvolveGCNO | 180s | 7.2s | 160s | 7.0s | 175s | 7.1s |
| TGCN | 165s | 6.9s | 145s | 6.7s | 155s | 6.8s |
| STWave | 155s | 6.5s | 135s | 6.3s | 145s | 6.4s |
| STAEformer | 190s | 7.5s | 170s | 7.3s | 200s | 7.4s |
| PDFormer | 210s | 7.8s | 195s | 7.6s | 220s | 7.7s |
| **STDMD** | **110s** | **4.2s** | **90s** | **4.0s** | **100s** | **4.1s** |

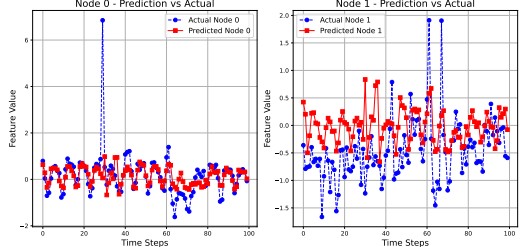

Figure 5: Comparison of actual vs. predicted node attributes.

feature dimensions, and $k_{\mathrm{spt}}$ and $k_{\mathrm{qry}}$ denote the sizes of the support and query sets, respectively. In contrast, baselines such as GConvGRU, STWave, and TGCN offer moderate efficiency, while DCRNN and EvolveGCNO incur higher costs due to more complex architectures. The heaviest models, STAEformer and PDFormer, exhibit the largest overhead, reflecting their larger parameterization. These results highlight STDMD's ability to balance accuracy and efficiency, making it particularly suitable for deployment in resource-constrained environments.

### 5.8 CASE STUDY

Figure 5 illustrates the comparison between actual and predicted feature values for two representative nodes over 100 time steps. The blue dashed lines correspond to the ground truth, while the red solid lines denote the predictions generated by STDMD. Overall, the model is able to accurately capture temporal patterns and fluctuations, demonstrating strong alignment with the observed trajectories. Nonetheless, its performance declines in scenarios with abrupt variations, such as sharp peaks and sudden drops, indicating limitations in temporal adaptability. Moreover, the close correspondence between the two nodes' feature dynamics highlights the model's ability to capture spatial dependencies within the graph. This case study underscores the robustness of STDMD for dynamic graph modeling, while also pointing to future opportunities for enhancing its capacity to handle rapid structural or temporal transitions.

## 6 CONCLUSIONS

In this paper, we introduced STDMD, a spatiotemporal dynamic meta-learning framework that enhances dynamic GCNs for node attribute forecasting. By innovatively constructing support and query sets, STDMD integrates spatial and temporal tasks into a unified optimization process, enabling effective capture of both structural dependencies and temporal dynamics. Extensive experiments on three real-world datasets demonstrate that STDMD consistently outperforms state-of-the-art baselines in terms of accuracy, efficiency, and robustness.

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
