# OpenReview forum: "Spatial-Temporal-Decoupled Meta Learning for Dynamic Node Attribute Forecasting"
_ICLR.cc/2026/Conference — ICLR 2026 Conference Withdrawn Submission_

### Official Review · Reviewer_CJhh · 2025-10-27

**Soundness:** 2
**Presentation:** 1
**Contribution:** 2
**Rating:** 2
**Confidence:** 3

**Summary:**

This study considers the dynamic node attribute forecasting task on discrete-time dynamic graphs (DTDGs). A new STDMD method was then proposed based on the meta-learning technique that can decouple and dynamically refine spatial and temporal representations through an iterative meta-optimization process. Experiments preliminarily demonstrate the effectivenss of STDMD.

**Strengths:**

**S1**. The key idea of applying the meta-learning technique, which decouple spatial and temporal information, to dynamic node attribute forecasting seems interesting.

**Weaknesses:**

**W1**. The overall presentation of this paper need further improvement.

In lines 30-38, there are no related citations for the description of possible applications of node attribute forecasting.

The font size of Fig. 2 is too small, which is hard to read.

This paper does not provide any appendies to describe further details (e.g., see **W2**-**W6**), which are usually necessary.


***
**W2**. There a several unclear statements with weak motivations, which need further clarification.

In the caption of Fig. 1, it was claimed that 'traditional models mainly rely on direct spatial connections' but what traditional models and why (i.e, due to what mechanims)? It is suggested to combine the toy example in Fig. 1 with a set of concerete traditional methods (e.g., baslines in experiments) with clear citations, which can better demonstrate the limitations of existing techniques.

In lines 121-124, it was claimed that 'most existing methods focus on macro-level spatiotemporal features, overlooking fine-grained dynamics in evolving node attributes'. However, it reamins unclear how to define and measure macro-level and fine-grained sptaio-temporal features (e.g., in terms of what).

In lines 157-161, it was claimed that the proposed method consider dyanmic graphs, while the sketch in Fig. 2 shows that STDMD only uses a static graph (snapshot) as input, which may not be consistent. Moreover, dynamic node attributes wer also not illustrated in Fig. 2.

For the problem statement of node attribute forecasting (i.e., lines 159-161), should the edge set $E_{t+1}$ of next time step be provided as the input when predicting attributes ${\bf{X}}_{t+1}$ of next time step. Otherwise, how does the node attributes forecasting task treat the variation of the lastest graph topology?

The definitin of dynamic graph in lines 189-192 is different from that in the formal problem statment (i.e., lines 157-159).

Many details of STDMD (e.g., how to constuct support and query sets of both tasks via sampling) were not clearly demonstrated in the overview shown in Fig. 2. As a result, it is still unclear how does STDMD extactly work according to Fig. 2. It is suggested to combined Fig. 2 with a toy running example (e.g., a small dynamic graph where nodes are denoted by unique indices) to demonstrate how does it work.

It seems that Eq. (7) shoule use the element-wise multiplication (i.e., Hadamard product), while the current version of this paper uses the standard matrix multiplication operation.


***
**W3**. There is no pseudo-code to summarize the overall training and inference procedures of STDMD.


***
**W4**. Experiments are too simple, which may not fully validate the effectiveness of STDMD.

While high scalability is one of the highlighted advantages of STDMD, all the datasets summarized in Table 1 are very small (i.e., in terms of the numbers of nodes, edegs, and time steps), which cannot demonstrate STDMD's high scalability. Some large datasets (e.g., with more than millions of nodes and time steps) are suggested to be added in experiments.

There are no descriptions to introduce details of dynamic attributes in each dataset (e.g., diemsnaionlity of attributes, physical meaning and value range of each dimension).

Effciency analysis in Section 5.7 only compares the runtime of a small part of baselines, where results of some other baselines (e.g., GCN, GConvLSTM, STGCN, STSGCN, MegaCRN, MetaDGE, and DMetaGCRN as shown in Table 2) are missing.

Although the complexity of STDMD was given Sectin 5.7, it remain unclear how to derive such a complexity by considering the overall training and inference procedures, as there are no detailed derivations for each step. It is also suggested to add further comparison with other baselines' complexities.


***
**W5**. This work did not anonymously provide its code to ensure the reproducibility of experiments.


***
**W6**. There are no discussions about the limitaions of this work and possible solutions as futurre research directions.

**Questions:**

See **W1**-**W6**.

**Q1**. According to the problem statement in Section 3, this study still considers the conventional data model of discrete-time dynamic graphs (DTDGs). Some SOTA temporal link prediction tehcniques [1-4] consider the more challenging yet realistic continusou-time dynamic graphs (CTDGs). Can the proposed method be extend to the more advanced CTDGs?

**Q2**. Another advantage of SOTA temporal link prediction / dynamic network embedding techniques is that they can support the advanced inductive inference, which can directly generalize the model trained on historical known topology to new unseen nodes. According to the problem statement in Section 3, this study still considers the conventional transductive setting. Can STDMD be extended to support the advanced inductive inference?

```
[1] Foundations and Modeling of Dynamic Networks Using Dynamic Graph Neural Networks: A Survey. IEEE Access 2021.
[2] A Survey on Embedding Dynamic Graphs. ACMM Computing Surveys 2021.
[3] Dynamic Network Embedding Survey. Neurocomputing 2022.
[4] Temporal Link Prediction: A Unified Framework, Taxonomy, and Review. ACM Computing Surveys 2023.

---

### Official Review · Reviewer_jePf · 2025-10-30

**Soundness:** 2
**Presentation:** 2
**Contribution:** 2
**Rating:** 4
**Confidence:** 5

**Summary:**

This paper focuses on dynamic graph forecasting and proposes STDMD, a dynamic graph neural network that integrates meta spatial-temporal decoupling to model evolving dependencies between nodes. The approach employs meta-learning to adapt spatial and temporal representations through iterative optimization, aiming to enhance prediction accuracy and robustness. Experimental results on real-world datasets show performance improvements.

**Strengths:**

1. The problem studied in this paper is important.

2. The experimental results are presented in a diverse manner.

**Weaknesses:**

1. The two challenges addressed in the paper—(i) difficulty in capturing complex spatiotemporal dependencies, and (ii) inability to adapt to evolving dynamic graph patterns—are not unique and have been extensively studied.

2. Since prior work has already applied meta-learning to dynamic graph tasks, the paper should clarify its main differences from these existing methods.

3. The temporal and spatial correlations are still modeled separately, similar to other related methods, and thus the proposed approach does not effectively solve the stated challenge that “conventional models often treat spatial and temporal information independently.” Meanwhile, this issue has already been well addressed in models such as STSGCN.

4. The datasets used are relatively small, and the rationale for their selection is unclear. Moreover, the datasets differ from those used in the compared baselines.

5. There is a misclassification in the description of baseline methods. DMetaGCRN does not involve meta-learning but is instead a meta graph generator by name.

6. Although the authors claim robustness for the proposed method, no experiments are provided to support this statement.

**Questions:**

Please see the weaknesses.

---

### Official Review · Reviewer_qmVa · 2025-10-31

**Soundness:** 3
**Presentation:** 3
**Contribution:** 3
**Rating:** 4
**Confidence:** 3

**Summary:**

This paper proposes STDMD, a dynamic GCN-based framework that decouples spatial and temporal tasks and applies a gradient-based meta-learning loop to adapt to evolving spatiotemporal patterns for node-attribute forecasting. The design includes (i) a GCN encoder with an LSTM-like gating update to produce dynamic node embeddings, (ii) task construction that samples support/query splits for both spatial (edge-based) and temporal (node/time-based) tasks, and (iii) dual-level meta-optimization that updates model parameters using both task losses. Experiments on three datasets show improvements over a wide set of baselines.

**Strengths:**

1. The decoupling of spatial/temporal tasks and then jointly meta-optimizing is intuitive, and the motivation examples are persuasive.
2. The model design is straightforward and implementable.
3. The experiments are extensive and persuasive enough to validate the effectiveness of the proposed method.

**Weaknesses:**

1. The novelty and contribution of this paper are unclear. This paper does not convincingly argue a principled novelty over prior meta-dynamic GNN works (MegaCRN, MetaDGE, DMetaGCRN, et al). The related work cites those, but does not clearly articulate what is conceptually new beyond packaging.
2. While decoupling is intuitive, the paper lacks deeper analysis showing when/why decoupling + meta-learning provably helps compared to a unified objective. Can multi-task MoE or multi-task prompting learning solve this problem?
3. Providing a theoretical discussion or analysis can better support the proposed method, which would substantially strengthen the depth of the work.
4. How does the method handle when contrastive spatial loss may conflate structure and attributes? The spatial objective forces embeddings of linked nodes to be similar; this is reasonable for some domains but not all (heterophilic graphs or cases where connected nodes have purposely different attributes).

**Questions:**

See weaknesses

---

### Official Review · Reviewer_agsz · 2025-11-01

**Soundness:** 1
**Presentation:** 3
**Contribution:** 2
**Rating:** 2
**Confidence:** 4

**Summary:**

This paper proposes STDMD, a meta-learning framework for dynamic node attribute forecasting. The method separates spatial and temporal learning tasks into independent meta-objectives and combines them through a weighted loss. Experiments on three real-world datasets demonstrate its effectiveness.

**Strengths:**

1. The idea of spatial–temporal decoupling in a meta-learning context is conceptually interesting and could improve interpretability.

2. The framework is simple, and clearly presented, making it easy to reproduce and extend.

3. Experimental results show consistent, albeit modest, improvements across three datasets.

**Weaknesses:**

1. The proposed method lacks theoretical grounding. The spatial–temporal decoupling lacks formal justification; the two losses are trained independently and combined via a fixed coefficient $\beta$ without analysis. Does the decoupling offer any real learning benefit, or is $\beta$ merely a heuristic weighting term?

2. The method remains shallow compared to recent dynamic graph learning frameworks (e.g., MetaDGE, DMetaGCRN). Specifically, STDMD operates on dynamic graphs but lacks an explicit mechanism to model or learn structural evolution. The spatial–temporal decoupling is implemented only by training two independent losses without any explicit learnable interaction between them.

3. The evaluation is limited to small-scale datasets. Can the model scale to larger graphs (such as windmilllarge)?

4. The authors have not released the source code and do not indicate any plan to do so, which limits the reproducibility of the work.

**Questions:**

See Weaknesses.

---

### Note · Authors · 2025-11-12

I have read and agree with the venue's withdrawal policy on behalf of myself and my co-authors.